# Development, Application and Evaluation of an Active Learning Methodology for Health Science Students, Oriented towards Equity and Cultural Diversity in the Treatment and Care of Geriatric Patients

**DOI:** 10.3390/ijerph192114573

**Published:** 2022-11-06

**Authors:** Manuel Sánchez De Miguel, Aintzane Orkaizagirre-Gomara, Andrea Izagirre-Otaegi, Francisco Javier Ortiz de Elguea-Díaz, Iker Badiola-Etxaburu, Ainara Gómez-Gastiasoro

**Affiliations:** 1Department of Basic Psychological Processes, Faculty of Psychology, University of the Basque Country, 20018 Donostia-San Sebastian, Spain; 2Biodonostia Health Research Institute, 20014 Donostia-San Sebastian, Spain; 3Department of Clinical Nursing-II, Faculty of Medicine and Nursing, University of the Basque Country, 20018 Donostia-San Sebastian, Spain; 4Donostia University Hospital, Osakidetza, Basque Health Service, 20018 Donostia-San Sebastian, Spain; 5Department of Cellular Biology and Histology (Dentistry), Faculty of Medicine and Nursing, University of the Basque Country, 48940 Leioa-Bilbao, Spain

**Keywords:** invisible care, geriatric patients, health science students, active learning methodologies, intervention program, sustainable development goals, 2030 Agenda, CCI-U questionnaire

## Abstract

The increased aging of populations and rises in immigration have prompted the design of new methodologies and instruments for fostering the invisible care of geriatric patients among health science students in accordance with the 2030 Agenda and the SDGs. A total of 656 psychology, nursing and dentistry students participated in this study, which had a pretest–posttest design and was implemented over the course of three academic years. The intervention groups received training using an active learning methodology based on a case study involving a geriatric patient; specifically, a Maghrebi woman. The control groups were not exposed to the case study. The CCI-U questionnaire was designed ad hoc to evaluate the acquisition of invisible competences for caring for geriatric patients in accordance with their age, sex, emotional situation and ethnic origin. The questionnaire had a reliability of α = 0.63 to 0.72 and its factor solution was found to have a good fit. Students in the intervention groups scored higher than those in the control groups, with the difference being statistically significant for ethnic origin in all three undergraduate courses and all three academic years. The proper application of this active learning methodology fosters the invisible care of geriatric patients among students in accordance with the 2030 Agenda.

## 1. Introduction

### 1.1. Aging and Multiculturalism

According to a report published by EUROSTAT [1], there are 3,459,090 immigrants aged 65 years or over currently living in Europe. The latest data indicate that 412,198 of them reside in Spain [2]. This is approximately 4% of the entire Spanish population aged 65 years and older (9.5 million).

The number of inhabitants in Spain (47.4 million) is expected to increase by 1 million over the coming 15 years [3]. Forecasts for 2035 predict that the +65 population will increase from 20% to 26.5% of the entire population. These figures warn of a gradual aging of the Spanish population, and it should also be borne in mind that the almost 1.3 million immigrants on the Spanish census who are currently aged between 45 and 64 years will be gradually joining this older demographic segment in the next decades [2].

It has been calculated that, in March 2020, at the start of the COVID-19 pandemic, 390,000 native Spaniards and immigrants were living in nursing homes in Spain. Furthermore, the number of people with dementia in Spain has risen, from 113,874 in 2011 to 443,976 in 2020 [4]. According to data published by the Alzheimer Europe Foundation, Spain is expected to have 1 million cases of dementia by 2025 and 1.7 million by 2050, representing 4% of the population forecast for that country by the middle of the century [5]. In 2011, 1.27% of the native population in Spain aged 65 years or older was diagnosed with Alzheimer’s disease. In the same age group, 0.43% of the immigrant population in Spain was diagnosed with Alzheimer’s. In 2019, 0.08% of the population in Spain < 65 years, 1.97% of those aged 65–79 years and 11.16% of those aged 80 years or over had diagnosed Alzheimer’s [4]. These indices indicate that the increased aging of the native and immigrant population could be linked to a higher number of cases of dementia. It should also be remembered that the 2030 Agenda for Sustainable Development contemplates an integrated agenda for everyone, with special emphasis on avoiding the exclusion and vulnerability of older adults [6].

### 1.2. Sustainable Development Goals (SDGs) and Invisible Care for Older Adults

Back in 2015, the United Nations proposed 17 different Sustainable Development Goals (SDGs) with the aim of enhancing economic, social and environmental sustainable development. These goals include ensuring healthy lives, promoting well-being for all at all ages and achieving gender equality and empowering all women and girls. In addition, the SDGs aim to ensure high-quality education, paying special attention to cultural diversity and gender equality [7]. To demarcate our intervention and objectives in a wider framework, the SDGs can be used to find solutions to the present issues with an interrelated and integrative approach.

However, new care demands oriented towards integrated care require us to think about multidisciplinary care [8,9] and to promote its presence in university health science degrees, expanding this training to also include other health professionals [10,11], such as physicians, psychologists and physiotherapists, among others.

The concept of care, which has a long tradition in nursing, began to be redefined at the beginning of the second decade of the current century (2010–2015). Ten years earlier, Colliere [12] had initiated this process of reflection by adopting a psychosocial approach to care, defining the most important type of care as that which is invisible and not recorded, referring to knowledge about the care recipient and their environment. Ausserhofer et al. [13] identified care related to patient comfort, communication with patients, the updating and development of care plans and the education of patients and their families as the types of care that European nurses most often do not provide due to unfavourable working conditions or low nurse/patient ratios. This results in the worsening of the quality of care provided, which in turn is reflected in a decrease in patient satisfaction, a drop in indicators [14] and even an increase in the mortality rate [15,16].

Various authors have studied the devaluation and invisibility of comprehensive care in nursing [17], and others [18] have identified specific care interventions, attitudes and behaviours that tend to go unnoticed or to which less value is attached in comparison with visible and quantifiable tasks, such as nursing techniques and tasks delegated by physicians. These authors define this type of care as ”invisible care”, since it involves interventions that are not recorded, transmitted to colleagues or valued institutionally, despite requiring much time and having a positive impact on the well-being, autonomy and safety of patients and their families.

In this same study [18], the authors identified the following as variables of invisible care: the education of patients and their families regarding the new situation and self-care, emotional support, comfort, pain reduction through non-pharmacological treatments and collaboration with health professionals to offer patient-centred care. In addition to being important to well-being and recovery, these types of care are also necessary elements within comprehensive care. However, they rarely enjoy the same degree of recognition as more technical and specialist types of care.

Invisible care is often more closely linked to people’s humanist and emotional dimensions, something which does not appear to be a coincidence and which supports the idea that the type of care that is more closely related to emotional aspects is less highly valued and relegated to second place in favour of the more technical aspects of caring for a patient. In this sense, Allen [19] explores the invisibility of organisational work in nursing, which has increased as a result of certain tasks being delegated to nursing staff by physicians but which should not be confused with invisible care during patient interactions.

In sum, the starting point for the present study was an acknowledgement of the importance of providing high-quality education (SDG 7) that fosters invisible care competences for personal interactions with patients oriented towards SDG 3 (ethical and humanist care of geriatric patients), SDG 5 (gender equality) and SDG 10 (reducing inequality stemming from age and/or cultural origin).

### 1.3. Active Learning Methodologies for Invisible Care and Its Evaluation

Invisible care is not always addressed in a sufficiently detailed manner in university training programmes. This type of care requires prior thought and reflection and forms the basis of future interactions with geriatric patients. In the last three years (2020–2022), our university has developed/carried out a total of 11 teaching innovation projects linked to Agenda 2030 and health science (four in nursing, one in medicine, five in psychology and one in pharmacy). However, our project is the only one developed at an interdisciplinary level in health science at this university.

The importance of active learning methodologies for health science studies lies in the way these methodologies link professional knowledge acquisition, research, interdisciplinary learning, clinical reasoning and patient-specific interventions [20].

The active learning methodology has the potential to link the three main domains of knowledge required by healthcare professionals: (1) clinical reasoning, assessment and profession-specific interventions; (2) fundamental sciences that underpin basic professional knowledge and its relationship to research; and (3) knowledge resulting from collaborative research between many professional groups. This knowledge is especially oriented to the acquisition of professional competencies and professional accreditation in the university environment [20].

Using an active learning methodology based on a case study of a female geriatric patient, we aim to foster individual and collective reflection among students in order to encourage the development of non-prejudiced, stereotype-free attitudes towards geriatric parents (see Figure 1) [21] and to promote interactions based on patients’ individual characteristics and needs, particularly in relation to their age [22,23], sex [23], emotional situation [18] and cultural origin [24,25], thereby enabling the practice of invisible care.

The implementation of the active learning methodology promotes reflection and a change in attitudes (see Figure 1) oriented to the invisible care of the geriatric patient. The change in attitudes must emerge prior to the progressive acquisition over time of the competencies and their fit with SDGs 3, 5 and 7.

In addition to technical training that results in the acquisition of professional competences, there is also space for fostering invisible care competences in a balanced manner oriented towards SDGs 3, 5 and 10. These competences are located within the framework of the professional–patient relationship and may emerge in a type of care that is more humanised, respectful and sensitive to constantly changing situations. In this sense, Darbyshire and McKenna [26] ask whether basic nursing care—we would add invisible care as well—has become a ”ghost in the machine”, since its absence is justified within syllabuses by the fact that it is considered to be implicit and dispersed into existing curricula; hence, the need to render invisible care visible. It is here that the use of active learning methodologies is justified. In our study, we used a case study method involving a patient (Shelima) and her clinical treatment. Using the problem-based learning through inquiry method (IBL) [27], our aim was to encourage students in the intervention groups to engage in reflective action and critical thinking [28] in order to become aware of this type of care and render it visible. By applying this innovative teaching process, we hoped to help students avoid the passive model of knowledge acquisition. The idea was that this new knowledge would emerge from new shared and innovative teaching experiences that would contribute to better academic performance by highlighting the usefulness of the subject for students’ future professional activities [29].

Finally, it is important to point out that this methodology involves a recalibration of the teaching scaffolding and the help provided to students through the IBL process. The aid required depends on academic level (first, second or third year), with more intense help and guidance being necessary at the start of the activity. As students move through their degree, however, the scaffolding provided should gradually decrease in order to bestow on them a greater degree of autonomy [30].

These circumstances justify the aim of the present study, which was to design and apply a new active learning methodology for the acquisition of invisible competences in accordance with the SDGs and to evaluate it pre- and posttest using a new instrument designed ad hoc (CCI-U). This strategy enabled us to measure the progress made by students in the acquisition of invisible competences beyond mere academic performance within the framework of the 2030 Agenda.

## 2. Materials and Methods

### 2.1. Participants

The participants were 656 students (85.2% women) from the faculties of nursing, psychology and dentistry (see Table 1). A total of 99.2% of the students were Spanish, 0.1% were from Morocco, 0.2% were from East Europe and 0.5% from South America. The percentage of immigrant students in the intervention groups was 0.15%. All were undergraduates over the course of three academic years (2019 to 2022) and enrolled in the following subjects: Psychology, Basic Nursing Methodologies and Quality and Safety in Nursing Care. A sample of 34 students of human histology (dentistry) were added in the last year of the study (2021–2022) in order to further analyse the effect of the intervention program on other health science students. This increase was undertaken after the major restrictions resulting from the COVID-19 pandemic were lifted.

A total of 576 students (87.8%) participated in the pre- and posttest phases; the remaining 80 (12.2%) failed to complete the posttest phase and were included in the sample only to ensure the sample size necessary for the factor analyses.

### 2.2. Instruments

Short Sociodemographic Questionnaire: data were collected regarding participants’ sex, academic level and degree course.

Invisible Care for Health Science Studies Questionnaire (*Cuestionario de cuidado invisibles en ciencias de la salud* (CCI-U)): This questionnaire was designed ad hoc and comprises 17 items (see Appendix A). It measures the acquisition of competences for invisible care in accordance with the patient’s age (factor 1: four items), sex (factor 2: four items), emotional situation (factor 3: three items) and ethnic-cultural origin (factor 4: six items). Of the 17 items, 13 (76%) are phrased as reverse items to avoid response bias. Items are rated on a six-point Likert-type scale (1—completely disagree, 2—mainly disagree, 3—partly disagree, 4—partly agree, 5—mainly agree and 6—completely agree). This six-point scale prevents centrality bias in the responses given.

### 2.3. Procedure

Five experts collaborated in the design of the CCI-U questionnaire, assessing and selecting the items proposed for the new instrument. Three of them were researchers and university professors specialized in geriatric nursing (invisible care), and another was a professor and researcher in neurosciences specialized in dementia and Alzheimer’s. The fifth expert was a professor and researcher specialized in attitudes and stereotypes towards the elderly and cross-cultural studies.

During this initial process, the Content Validity Index (CVI) indicator was used. Members of the research team proposed a total of 100 items for measuring invisible care in accordance with a patient’s age, sex, emotional situation and ethnic origin. The proposal of these 100 items was based on a review of the literature and pre-existing scales, placing the health science student and the geriatric immigrant patient at the centre of the questionnaire and making sure to represent the four main areas of invisible care: the gender, age, emotional state and ethnicity of the geriatric patient. The items were not extracted from other validated tools and were based on the most common experiences that occur in geriatric centres with immigrant patients and on the experience of the professionals who work in these centres with immigrant elderly people.

The Content Validity Index (CVI) was calculated based on the experts’ scores for all items, using a four-point Likert-type scale (1 = not relevant, 2 = needs serious revision, 3 = relevant but needs minor revision, 4 = quite relevant). The CVI values were greater than 0.79 for 32 items, with scores of between 3 and 4 on the Likert-type scale, thus indicating good content validity [31].

To establish an English-language version of the CCI-U scale, the items were translated by an expert translator with extensive experience in the academic field in general and in psychology in particular [32]. The final English version (see Appendix A) was then translated back into Spanish [33] by three other university faculty members working independently from those responsible for the initial adaptation proposed. Finally, a definitive consensus model (Kappa = 0.89, *p* < 0.03) was agreed on for the final version of the CCI-U scale. The consensus model (Kappa = 0.93, *p* < 0.01) was then agreed on for the Spanish version of the CCI-U. Finally, the CCI-U questionnaire was administered to a small sample of 15 participants using a “face validity procedure” to test for possible mistakes and general understanding. The final version presented no problems.

The G*Power program (see 3.1.9.6, Franz Paul: Kiel, Germany) was used to calculate the minimum sample size required in light of the study design and the use of Student’s *t*-tests (*n* = 2 groups, control vs. intervention), with a medium effect size (0.50 to 0.79), a confidence level set to 95% and power = 0.8. The results indicated that the minimum sample size required per academic year was 128 [34].

### 2.4. The Intervention Program

The students in the intervention group participated in the active learning methodology in class time, during practical sessions held from weeks 4 to 11 of the 15 week term (see Figure 2). The methodology was applied over the course of seven face-to-face sessions in the classroom, each lasting 2 h. All the material included in the active learning methodology was uploaded to the e-gela platform (Moodle). The working reports drafted by the intervention groups for the case study (patient Shelima) were submitted to the faculty in paper format.

To evaluate the efficacy of the active learning methodology intervention programme, the CCI-U questionnaire was activated on the e-gela (Moodle) platform during the pretest (week 6) and posttest (week 10) phases. Both the control and intervention groups had access to the questionnaire. Following the pretest measures, the intervention group attended a training seminar on invisible care (week 7). As a complementary training activity, outside of face-to-face class time, students of the intervention group were provided with access to a video link on the e-gela platform (Moodle) where they could watch two documentaries, one about the French psychiatry revolution led by Philippe Pinel (1745–1826) and one about the current functioning of short-stay centres for patients with mental disorders. After watching the videos, these students were asked to respond to a ten-question quiz (Socrative Pro) about the most important contents linked to the historical evolution of care.

Throughout weeks 7, 8 and 9 (see Figure 2), students from the intervention group worked on the case study involving the geriatric patient Shelima from the Maghreb. The storyboard of the case encompassed four principal scenarios in a hospital context and one in a nursing home context. The first scene described a conflictive situation in a work therapy session, where Shelima was rejected by the other patients due to being a woman and from the Maghreb. The second scenario featured a conflict between Shelima and the canteen staff over the midday meal menu offered during Ramadan, as well as over some of the patient’s cultural beliefs that prompted her to reject both the menu and her mediation. The third scenario was set during afternoon visiting hours, when Shelima was visited by a friend who lied to her about a series of family events, since she believed that, due to her age, she should not be informed of certain occurrences. Finally, the fourth scenario placed Shelima in a delicate emotional situation when she learned that her daughter did not want to visit her or have anything to do with her. In all the scenarios of this IBL, students from the intervention group were asked to articulate solutions using invisible care and outline them in a case method report.

At the end of the session held in week 9, all the solutions proposed by each intervention group were shared during a cooperative debate.

During week 11, after the posttest evaluation (CCI-U), individual student (control and intervention) results were published (using an anonymous four-digit identifier) and each working group (intervention) was provided with feedback on the case method report submitted.

Unlike the intervention group, the control group was not exposed to Shelima’s case study (IBL) during weeks 7, 8, 9 and 10. In this time, the control group received teaching in another subject. The two groups received the same teaching and subjects without an active learning methodology during the rest of the weeks.

### 2.5. Data Analyses

All data were processed using the SPSS program (see 22.0, IBM: Armonk, NY, USA). Student’s *t* test was used to determine whether or not significant differences existed between the control and intervention groups in terms of their pretest and posttest scores. The Wilcoxon *U* test was performed to determine the degree of individual improvement among students following the application of the active learning methodology. The AMOS program (see 22.0, IBM: Chicago, IL, USA) was used for the confirmatory factor analysis.

### 2.6. Ethical Considerations

The study was approved by the ethics committee of the University of the Basque Country (CEISH num. M10_2021_278) prior to data collection. All participants received and signed an informed consent document. All the information collected was registered anonymously. Refusal to participate in the study had no impact on students’ academic grades.

## 3. Results

### 3.1. Factor Analysis and Reliability

An exploratory factor analysis (EFA) of the CCI-U theoretical construct was conducted with a sample of 335 students using the principal components method with varimax rotation. A total of 15 items with loadings of less than 0.30 were removed from the initial version. The Kaiser–Meyer–Olkin (KMO) index, which was used as a measure of sampling adequacy, was 0.87. Bartlett’s test was statistically significant [χ^2^ (210) = 1725.83, *p* < 0.001], enabling the inclusion of the factors selected in the factor analysis. Factor loadings were between 0.38 and 0.76. The four-dimensional solution with a total of 17 items (see Table 2) explained 47.27% of the variance of the CCI-U.

Next, a confirmatory factor analysis (CFA) was carried out with a different subsample (*n* = 321) to test the goodness of fit of the CCI-U with a four-factor structure (17 items). The kurtosis and asymmetry indicators were all less than 3, thereby indicating univariate normal distribution.

The goodness of fit of the CCI-U questionnaire was assessed using the following indicators: (a) the ratio between chi squared and degrees of freedom (χ^2^/*df*); (b) the comparative fit index (CFI); (c) the incremental fit index (IFI); and (d) the root mean square of approximation (RMSEA).

Hu and Bentler [35] consider CFI and IFI values of above 0.90 to be acceptable, and according to Marsh et al. [36], values of between 0.05 and 0.10 are acceptable for the RMSEA. For the theoretical learning subscale, the following CFA values were obtained (*n* = 321): χ^2^ (113) = 275.24, *p* < 0.001; χ^2^/*df* = 2.436; CFI = 0.91; IFI = 0.90; RMSEA = 0.06. All items had saturations of over 0.30 (see Figure 3).

The analysis of the CCI-U revealed reliability coefficients of between α = 0.62 and 0.72 for the different factors. Four factors were found to have acceptable reliability values [37].

### 3.2. Intervention Effects on Outcome Variables

The control and intervention groups from the psychology, nursing and dentistry undergraduate courses were compared statistically to verify sample normality in terms of previous invisible care. The K-S test revealed a normal distribution (*p* > 0.05) in all three samples. The skewness values for all groups (Sk = −1.75 to 0.74) were less than 3 and kurtosis values (k = −0.89 to 4.01) were less than 8, confirming good symmetry and normality [38] prior to the intervention programme in each academic year. The results of the student analysis test (see Table 3) revealed, with a reliability level of 95% (*p* > 0.05), homogeneity between the control and intervention groups prior to the intervention programme.

The results returned by the Student’s *t*-test in the posttest phase after the implementation of the methodology to foster invisible care in accordance with patients’ ethnic and cultural origin (see Table 4) revealed a statistically significant group-level effect in psychology and dentistry, whereas in nursing no effect of the intervention on the subject was observed at a higher level (third year).

In relation to the evolution of the improvement achieved by the active learning methodology across the three academic years analysed, students in the intervention group on the psychology course obtained the best results in all four factors (F1 to F4), as measured in terms of invisible competences during the last academic year (2021–2022). The effect size for care in accordance with the patient’s ethnic origin was medium–large (D_z_ = 0.36 to 0.84) in this last academic year.

Among the students in the intervention group undertaking the nursing degree, a significant group improvement was observed during the last academic year (2021–2022) in the Basic Nursing Methodologies subject; in this case, in three of the four factors (F2 to F4) with a small effect size (D_z_ = 0.32 to 0.39). The other subject, Quality and Safety in Nursing Care, could not be analysed over time since it was only taught during the last academic year. Nevertheless, in this subject, significant group-level differences were observed only in F3 (emotional situation), although with a very large effect size (D_z_ = 1.12). As mentioned in the description of the participants, the results obtained by the first-year dental students could only be analysed in the last year of this project.

Significant group-level differences were observed only in F4 (ethnic origin) and had a small effect size (D_z_ = 0.36).

In order to explore individual differences in the psychology, nursing and dentistry degree intervention groups, we conducted a Wilcoxon test (see Table 5). Significant differences (*p* < 0.03) were found among psychology students for the invisible care in accordance with the patient’s ethnic origin factor (F3) over the three academic years studied. Statistically significant individual improvements (*p* < 0.05) were also observed for the invisible care in accordance with the patient’s sex factor (F2) in the three psychology intervention subgroups during the final academic year (2021–2022).

Among the students undertaking the nursing degree, no statistically significant individual improvements were observed in the Basic Nursing Methodologies subject during the 2020–2021 academic year. The greatest statistically significant improvement (*p* < 0.05) was found during the last academic year (2021–2022) in all factors except F3 (care in accordance with emotional situation); in contrast, among those taking the Quality and Safety in Nursing Care subject, the greatest statistically significant individual improvement (*p* < 0.05) was found in factors F3 (emotional situation) and F4 (ethnic origin).

Among those studying dentistry, greater statistically significant individual improvements (*p* < 0.03) were observed in all factors except F2 (patient’s sex).

## 4. Discussion

The present study analysed the effects of an active learning methodology applied to health science students through an intervention and educational innovation programme located within the framework of the SDGs [7]. The psychology, nursing and dentistry students who participated in the intervention groups scored higher for the acquisition of invisible competences for the care of geriatric patients than their counterparts in the control groups. These findings enabled us to fulfil our aim of developing, applying and evaluating an active learning methodology for acquiring invisible competences in geriatric care.

The CCI-U questionnaire, which was designed ad hoc to evaluate the acquisition of invisible competences by students, was found to have a correct factor structure, as ratified by good fit values in the confirmatory factor analysis. This new instrument was also found to have reliability values similar to those reported for the Perception of Invisible Nursing Care—Hospitalisation (PINC-H) questionnaire [39]. However, unlike the PINC-H, which reflects the perceptions of young and adult oncological patients about invisible care [39], and the Care-Q Caring Assessment instrument [40], which measures the satisfaction of all type of patients regarding two specific elements of invisible care, such as the comfort and confidence provided by the health care professionals, our results confirmed the good functioning of the CCI-U in the evaluation of an intervention designed to foster the acquisition of invisible competences.

The two instruments mentioned above (PINC-H and Care-Q) are not designed to evaluate care based on the perceptions of health professionals, nor are they meant to be applied in university training. In this sense, the CCI-U also performs better than the CIBISA scale [41], which is applied with nursing students and measures three principal factors (well-being, clinical safety and autonomy) but does not address changes in attitude prior to the acquisition of competences in invisible care. From the perspective of teaching innovation, the CCI-U scale focuses on the invisible care of geriatric patients and does so in accordance with the 2030 Agenda, seeking to explore the “bridge” that exists between non-stereotypical attitudes and invisible care competences (see Figure 1). It also includes other health professions (psychology, medicine, dentistry, etc.) as well as nurses and fits in well with the application of active learning methodologies.

The results of the study reveal that the active learning methodology used promotes greater group-level awareness among intervention groups regarding the invisible care of geriatric patients, particularly in relation to factors F3 (emotional situation) and F4 (ethnic origin). Improvements in F1 (patient’s age) and F2 (patient’s sex) were also observed following the intervention, although they were not as statistically significant. In terms of proximal attitudes to ageism and sexism, the results of our study are not consistent with those reported by other studies carried out in relation to geriatric care [23,42,43]. With regard to the nursing students in the intervention groups, the absence of any improvement in factors F1 and F2 may be explained by the fact that care itself is traditionally implicit in their chosen profession [44]. It may also be explained by a greater social awareness and responsibility in relation to the age and sex of geriatric patients, as indeed demonstrated by the high pretest scores for the invisible competences linked to these two factors obtained by the control groups for all three health science degrees.

In the specific cases of the intervention groups from the Psychology and Basic Nursing Methodologies courses, we were able to compare scores across three different academic years, observing a gradual improvement at a group level in the implementation of the active learning methodology, which was particularly reflected in the results obtained for the 2021–2022 academic year. The poorest implementation for all four factors analysed during the three different academic years was in 2020–2021. This may be explained by the restrictions and reduced interactions imposed following the initial lockdown of the university population (including health science students) during the COVID-19 pandemic for the first months of face-to-face teaching [45]. The pandemic suspended synchronous and face-to-face teaching, which is particularly beneficial for the cooperative work required by IBL and knowledge sharing from critical thinking, especially for teaching students in the early grades [46].

In relation to factors F3 (emotional situation) and F4 (ethnic origin), the results may plausibly be explained by the fact that, unlike age and sex, these are not strictly biological factors. Moreover, age- and sex-based stereotypes and discrimination in geriatric care are more commonly known and are therefore more susceptible to social desirability bias among respondents. When we address the emotional situation (F3) of a geriatric patient, we often do so from the perspective of caring for people with dementia, and our results, which revealed an improvement in pretest scores, may be explained by the initial presence of attitudes that were not particularly empathetic [23], avoidant behaviours among caregivers in response to poor prognoses and the irreversible nature of the situation [47,48] and the lack of specialist training in geriatric care among health science students [49].

It should be highlighted that invisible care in accordance with the patient’s ethnic origin (F4) was the factor that responded best to the active learning methodology and the intervention programme with health science students. The low percentage of immigrant students in the intervention groups was not expected to affect the results.

This margin for improvement is consistent with that reported by other studies regarding the existence of racist prejudice towards patients [50,51,52,53,54]. Another possible explanation may be found in the emphasis placed on the Maghrebi origin of the geriatric patient Shelima in the four principal scenarios of the case study storyboard. Everything seems to indicate that age and sex intersect [23] with the ethnic factor, which emerged most noticeably in the scenarios portraying the conflict around the group’s acceptance of Shelima during a work therapy session, the conflict regarding diet during Ramadan, the patient’s poor social–family network and her inability to cope with emotions.

In relation to the individual analysis performed using the Wilcoxon *U* test, the individual improvement observed in the dentistry intervention groups was greater than the group improvement observed using Student’s *t* test. This improvement encompassed all factors except F2 (sex). A similar pattern of individual improvement was observed in the psychology intervention groups, particularly in the 2021–2022 academic year, in which the group improvement was negligible. These results may be explained by the fact that the Wilcoxon *U* test is a non-parametric test and is therefore more sensitive when groups contain fewer than 30 subjects and have non-normal distributions. It is important to highlight that, again, ethnic origin (F4) was the factor most sensitive to individual student improvement, similarly to the observation in relation to group improvement.

One could argue that ethnic origin (F4) may be considered a biological factor and that it should have been analysed from this perspective within our active learning methodology. However, consistently with the way in which the invisible care of a Maghrebi geriatric patient was approached in this study, the four scenarios of the storyboard focused more on the patient’s social–cultural characteristics (identity, tradition, religious beliefs and cultural practices) than on her biogenetic make-up linked to the concept of race. We therefore opted to view the variable “ethnic origin” in accordance with recent contributions to the field of health science [55,56].

Furthermore, the use of this methodology and its evaluation using the CCI-U questionnaire enabled us to identify specific areas of improvement in certain groups. This was the case, for example, with the Quality and Safety in Nursing Care subject and the invisible care of geriatric patients in accordance with their sex (F2). In this case, a sharp drop (<5 points) was observed in this invisible competence following the intervention and application of the active learning methodology in the 2021–2022 academic year. Since this was the nursing group with the most practical experience (third year of the degree course), it is possible that contents from another subject [57] or real experiences with immigrant women in hospital practicums [58] may have interfered with the methodology.

The aim of this study was not to compare IBL with traditional teaching but to improve the design and efficacy of IBL within the active learning methodology in order to achieve greater acquisition of competencies in the four areas of invisible care. After this experience, we will in the future directly apply the IBL method to all students without control groups. We understand that IBL promotes greater critical thinking [28] by giving greater visibility to invisible care in line with the SDGs. The present study has certain limitations. First, only 50% of the subjects were analysed across all three academic years, which precluded a longitudinal overview of the implementation of invisible care across all groups. Second, of the 12 groups analysed, only 3 had more than 30 participants and 4 had 20 or fewer. This lack of balance may have affected the results obtained in the Student’s *t* tests. Third, the groups analysed were studying subjects from different academic levels. This may have resulted in higher invisible care scores among students in later years.

Future research should strive to include native male geriatric patients in this type of active learning methodology in order to enable comparisons with their immigrant female counterparts. It is also important to verify the stability of the new CCI-U questionnaire in other countries and cultural contexts. Finally, a social desirability scale should also be included in order to control for this variable.

## 5. Conclusions

From the perspective of teaching innovation, the new CCI-U scale presented here focuses on the invisible care of geriatric patients in accordance with the precepts of the 2030 Agenda, exploring the “bridge” that exists between non-stereotypical attitudes and competences linked to invisible care. This active learning methodology is also suitable for use in other health professions (psychology, medicine, dentistry, etc.) as well as nursing and fits in well with the application of active learning methodologies in general.

The good functioning of the CCI-U and the active learning methodology applied has important implications in the field of health science since they can foster the invisible care of geriatric patients from university training onwards alongside its more technical counterpart, rendering this type of care visible. We are currently adapting the active learning methodology for invisible geriatric care to incorporate multidisciplinary care [8,9] and interdisciplinary teaching (IDT) [59], integrating students from nursing, psychology and dentistry degree courses into the intervention groups.

## Figures and Tables

**Figure 1 ijerph-19-14573-f001:**
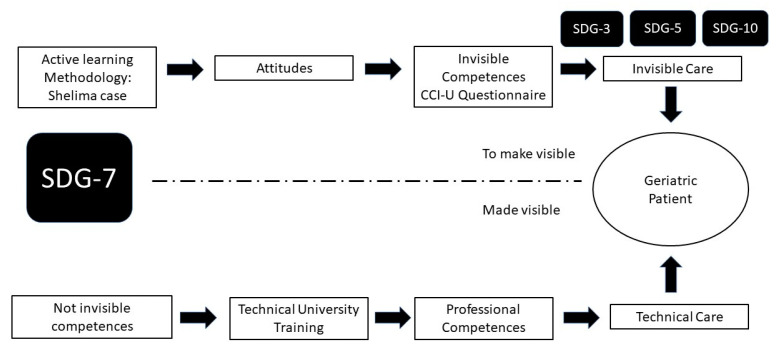
Invisible care vs. technical care.

**Figure 2 ijerph-19-14573-f002:**
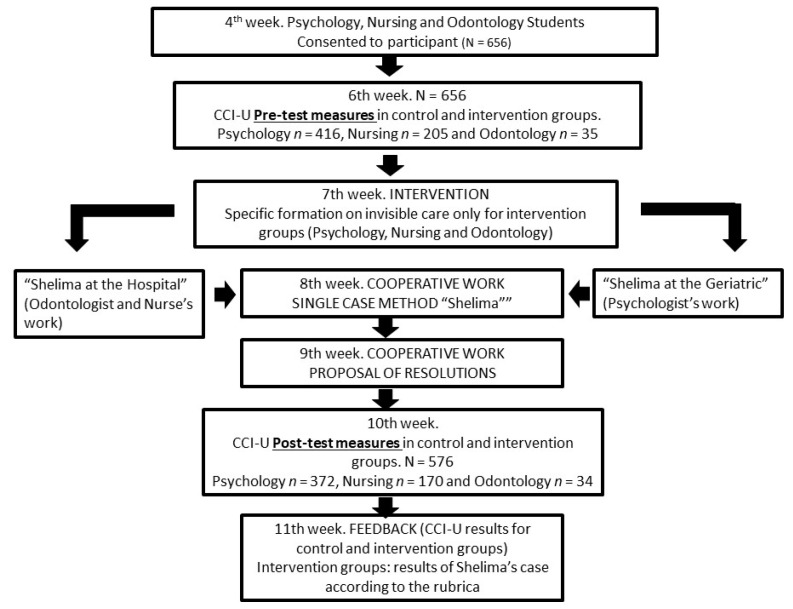
Flowchart of the active learning methodology on invisible care.

**Figure 3 ijerph-19-14573-f003:**
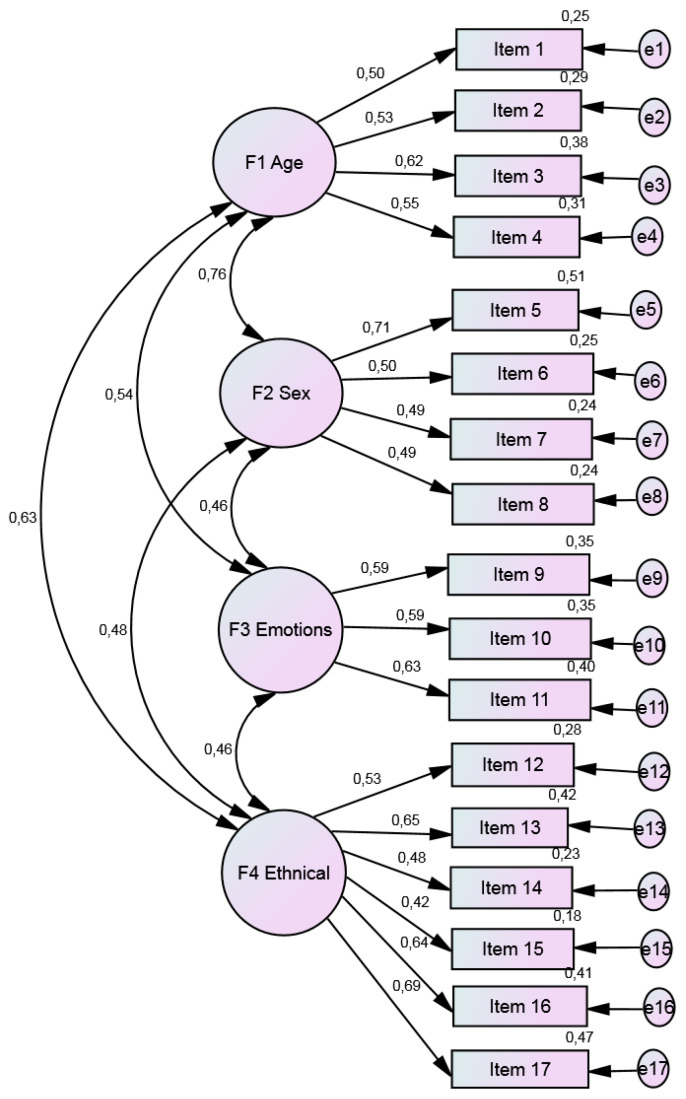
Confirmatory factor analysis of the CCI-U.

**Table 1 ijerph-19-14573-t001:** Sociodemographic data.

Control Groups							Intervention Groups				Age		
Academic Year and Degree	*n*	Female	%	Male	%	*n*	Female	%	Male	%	Range	M	SD
2019–2020 academic year													
Psychology	68	51	75.55	17	24.45	66	56	84.85	10	15.15	[17–43]	18.33	(2.51)
Nursing	21	20	94.52	1	5.48	20	15	75.00	5	25.00	[18–55]	20.95	(6.03)
2020–2021 academic year													
Psychology	66	57	86.40	9	13.60	66	56	85.10	10	14.90	[17–31]	18.07	(1.69)
Nursing	22	19	88.60	3	11.40	23	19	82.30	4	17.70	[18–48]	20.93	(5.30)
2021–2022 academic year													
Psychology	53	49	86.59	4	13.41	53	46	87.73	7	12.27	[17–47]	18.74	(3.69)
Nursing	42	38	90.69	4	9.31	42	34	80.43	8	19.57	[18–55]	21.27	(7.35)
Dentistry	17	11	66.66	6	33.14	17	16	94.11	1	5.89	[18–31]	19.37	(2.59)

**Table 2 ijerph-19-14573-t002:** CCI-U questionnaire. Exploratory factor analysis, descriptive statistics and reliabilities (*n* = 335).

	F1	F2	F3	F4			
Items	Factor Loading	Factor Loading	Factor Loading	Factor Loading	M (SD)	Asymmetry	Kurtosis
Item 1	0.459				5.65 (0.78)	−1.295	0.809
Item 2	0.440				4.96 (1.25)	−1.060	0.162
Item 3	0.543				4.91 (1.26)	−1.168	0.709
Item 4	0.385				5.16 (1.13)	−1.512	1.913
Item 5		0.641			5.34 (1.04)	−1.273	0.764
Item 6		0.691			4.93 (1.32)	−1.167	0.497
Item 7		0.567			5.54 (1.02)	−1.772	2.631
Item 8		0.523			4.70 (1.27)	−0.761	0.270
Item 9			0.508		5.63 (0.76)	−1.865	2.783
Item 10			0.745		4.76 (1.30)	−0.931	0.041
Item 11			0.766		4.65 (1.26)	−1.144	0.805
Item 12				0.596	4.74 (1.14)	0.818	0.615
Item 13				0.545	5.21 (1.01)	−1.567	2.347
Item 14				0.604	4.93 (1.49)	−1.432	0.961
Item 15				0.603	4.46 (1.40)	−0.638	0.341
Item 16				0.686	5.28 (0.99)	−1.707	2.401
Item 17				0.701	5.26 (1.01)	−1.544	2.372

Variance Explained 7.17% 8.51% 5.62% 25.96%; Reliabilities α = .63 α = .62 α = .65 α = .72; Range [1 to 6 points]; F1: Competences for treatment in accordance with patient’s age. F2: Competences for treatment in accordance with patient’s sex. F3: Competences for treatment in accordance with patient’s emotional situation. F4: Competences for treatment in accordance with patient’s ethnic-cultural origin.

**Table 3 ijerph-19-14573-t003:** CCI-U questionnaire. Pretest scores for psychology, nursing and dentistry students in the control and intervention groups, classified by academic year.

	Control		Intervention					
Subject, Year and Scale	M (SD)	*n*	M (SD)	*n*	t	df	*p*	Dz Cohen
Psychology								
2019–2020 academic year								
Group 1								
F1	4.66 (0.48)	45	4.51 (0.48)	44	1.367	87	n.s.	
F2	4.78 (0.69)	45	4.75 (0.61)	44	0.271	87	n.s.	
F3	4.99 (0.40)	45	4.94 (0.51)	44	0.537	87	n.s.	
F4	4.38 (0.50)	45	4.45 (0.67)	44	0.584	87	n.s.	
Group 2								
F1	4.61 (0.52)	23	4.77 (0.49)	22	−1.049	43	n.s.	
F2	4.73 (0.64)	23	4.62 (0.62)	22	0.543	43	n.s.	
F3	4.96 (0.42)	23	5.02 (0.49)	22	−0.465	43	n.s.	
F4	4.41 (0.46)	23	4.39 (0.51)	22	0.137	43	n.s.	
2020–2021 academic year								
Group 1								
F1	4.67 (0.51)	33	4.75 (0.51)	33	−0.560	64	n.s.	
F2	4.78 (0.62)	33	4.71 (0.53)	33	0.487	64	n.s.	
F3	5.07 (0.35)	33	5.14 (0.39)	33	−0.706	64	n.s.	
F4	4.39 (0.51)	33	4.54 (0.61)	33	−1.034	64	n.s.	
Group 2								
F1	4.67 (0.47)	33	4.58 (0.41)	33	0.828	64	n.s.	
F2	4.70 (0.77)	33	4.51 (0.63)	33	1.097	64	n.s.	
F3	5.05 (0.37)	33	4.89 (0.38)	33	1.690	64	n.s.	
F4	4.38 (0.51)	33	4.31 (0.70)	33	0.538	64	n.s.	
2021–2022 academic year								
Group 1								
F1	4.40 (0.42)	22	4.48 (0.53)	22	−0.606	42	n.s.	
F2	4.62 (0.46)	22	4.47 (0.62)	22	0.905	42	n.s.	
F3	4.71 (0.41)	22	4.94 (0.56)	22	−1.583	42	n.s.	
F4	4.29 (0.52)	22	4.40 (0.53)	22	−0.731	42	n.s.	
Group 2								
F1:	4.56 (0.36)	14	4.57 (0.36)	14	−0.035	26	n.s.	
F2	4.63 (0.45)	14	4.88 (0.79)	14	−1.061	26	n.s.	
F3	4.71 (0.48)	14	5.04 (0.45)	14	−1.928	26	n.s.	
F4	4.18 (0.50)	14	4.38 (0.85)	14	−0.777	26	n.s.	
Group 3								
F1	4.45 (0.44)	17	4.54 (0.31)	17	−0.718	32	n.s.	
F2	4.63 (0.50)	17	4.65 (0.54)	17	−0.167	32	n.s.	
F3	4.73 (0.44)	17	4.83 (0.49)	17	−0.733	32	n.s.	
F4	4.26 (0.57)	17	4.72 (0.42)	17	−1.548	32	n.s.	
Basic Nursing Methodologies								
2019–2020 academic year								
F1	5.16 (0.35)	21	5.16 (0.37)	20	0.354	39	n.s.	
F2	4.97 (0.65)	21	4.81 (0.68)	20	−0.150	39	n.s.	
F3	4.51 (0.33)	21	4.74 (0.74)	20	−1.641	39	n.s.	
F4	4.46 (0.25)	21	4.75 (0.41)	20	−1.795	39	n.s.	
2020–2021 academic year								
F1	5.04 (0.47)	22	5.08 (0.36)	23	−0.823	43	n.s.	
F2	4.72 (0.47)	22	4.87 (0.61)	23	−0.910	43	n.s.	
F3	5.02 (0.42)	22	5.34 (0.54)	23	−2.171	43	0.035	0.331 ^a^
F4	4.89 (0.44)	22	5.08 (0.55)	23	−1.279	43	n.s.	
2021–2022 academic year								
F1	4.83 (0.47)	28	4.79 (0.47)	28	0.354	54	n.s.	
F2	4.77 (0.47)	28	4.80 (0.63)	28	−0.150	54	n.s.	
F3	5.17 (0.53)	28	5.38 (0.53)	28	−0.717	54	n.s.	
F4	4.81 (0.65)	28	5.11 (0.60)	28	−1.809	54	n.s.	
Quality and Safety in Nursing Care								
2021–2022 academic year								
F1	4.88 (0.51)	14	5.04 (0.27)	14	−1.024	26	n.s.	
F2	4.93 (0.62)	14	4.82 (0.68)	14	0.477	26	n.s.	
F3	5.05 (0.41)	14	5.33 (0.42)	14	−2.104	26	n.s.	
F4	4.61 (0.66)	14	4.88 (0.47)	14	−1.252	26	n.s.	
Human Histology (Dentistry)								
2021–2022 academic year								
F1	4.56 (0.47)	20	4.40 (0.64)	17	0.823	35	n.s.	
F2	4.88 (0.46)	20	4.63 (0.47)	17	1.660	35	n.s.	
F3	5.12 (0.61)	20	5.00 (0.51)	17	0.667	35	n.s.	
F4	3.95 (0.49)	20	4.00 (0.57)	17	−0.331	35	n.s.	

^a^ Low Dz Cohen values indicate no significant difference; Range [1 to 6 points]; F1: Competences for treatment in accordance with patient’s age. F2: Competences for treatment in accordance with patient’s sex. F3: Competences for treatment in accordance with patient’s emotional situation. F4: Competences for treatment in accordance with patient’s ethnic-cultural origin.

**Table 4 ijerph-19-14573-t004:** CCI-U questionnaire. Posttest scores for psychology, nursing and dentistry students in the control and intervention groups, classified by academic year.

	Control		Intervention					
Subject, Year and Scale	M (SD)	*n*	M (SD)	*n*	t	df	*p*	Dz Cohen
Psychology								
2019–2020 academic year								
Group 1								
F1	4.78 (0.53)	45	4.72 (0.56)	44	0.652	87	n.s	
F2	4.78 (0.76)	45	4.90 (0.60)	44	−0.765	87	n.s	
F3	5.07 (0.54)	45	5.10 (0.63)	44	−0.202	87	n.s	
F4	4.37 (0.60)	45	4.69 (0.61)	44	−2.441	87	0.017	0.367
Group 2								
F1	4.76 (0.52)	23	5.12 (0.39)	22	−2.650	64	0.011	0.404
F2	4.74 (0.77)	23	5.13 (0.66)	22	−1.795	64	n.s	
F3	4.97 (0.61)	23	5.31 (0.40)	22	−2.172	64	0.036	0.331
F4	4.29 (0.57)	23	4.87 (0.55)	22	−3.272	64	0.001	0.450
2020–2021 academic year								
Group 1								
F1	4.83 (0.48)	33	5.07 (0.65)	33	−1.683	64	n.s	
F2	4.89 (0.73)	33	5.00 (0.51)	33	−0.669	64	n.s	
F3	5.06 (0.57)	33	5.20 (0.59)	33	−0.973	64	n.s	
F4	4.39 (0.61)	33	4.94 (0.59)	33	−3.709	64	0.001	0.463
Group 2								
F1	4.74 (0.57)	33	4.75 (0.44)	33	−0.010	64	n.s	
F2	4.67 (0.80)	33	4.68 (0.60)	33	−0.025	64	n.s	
F3	5.16 (0.48)	33	5.26 (0.45)	33	−0.863	64	n.s	
F4	4.39 (0.65)	33	4.72 (0.61)	33	−2.154	64	0.035	0.269
2021–2022 academic year								
Group 1								
F1	4.27 (0.60)	22	4.45 (0.59)	22	−0.982	42	n.s	
F2	4.45 (0.71)	22	4.82 (0.75)	22	−1.699	42	n.s	
F3	4.63 (0.78)	22	5.13 (0.43)	22	−2.624	42	0.012	0.404
F4	4.24 (0.64)	22	4.63 (0.53)	22	−2.191	42	0.034	0.631
Group 2								
F1	4.37 (0.69)	14	4.96 (0.39)	14	−2.779	26	0.010	0.544
F2	4.35 (0.82)	14	5.26 (0.79)	14	−3.341	26	0.003	0.655
F3	4.57 (0.93)	14	5.48 (0.28)	14	−3.508	26	0.002	0.687
F4	4.24 (0.77)	14	5.23 (0.39)	14	−4.293	26	0.001	0.841
Group 3								
F1	4.29 (0.65)	17	4.69 (0.48)	17	−2.055	32	0.048	0.363
F2	4.44 (0.76)	17	4.94 (0.62)	17	−2.098	32	0.044	0.371
F3	4.65 (0.87)	17	4.94 (0.64)	17	−1.091	32	n.s	
F4	4.24 (0.71)	17	4.85 (0.43)	17	−3.027	32	0.005	0.534
Basic Nursing Methodologies								
2019–2020 academic year								
F1	5.25 (0.31)	21	5.14 (0.67)	20	0.758	39	n.s	
F2	4.98 (0.51)	21	5.06 (0.45)	20	−0.516	39	n.s	
F3	4.02 (0.40)	21	4.85 (0.40)	20	−6.641	39	0.001	1.063
F4	4.17 (0.27)	21	5.02 (0.27)	20	−7.419	39	0.001	1.187
2020–2021 academic year								
F1	5.19 (0.42)	22	5.16 (0.56)	23	0.225	43	n.s	
F2	4.84 (0.57)	22	5.09 (0.57)	23	−1.564	43	n.s	
F3	5.12 (0.46)	22	5.52 (0.40)	23	−3.124	43	0.003	0.476
F4	4.89 (0.39)	22	5.04 (0.64)	23	−0.958	43	n.s	
2021–2022 academic year								
F1	4.85 (0.47)	28	5.04 (0.45)	28	−1.353	52	n.s	
F2	4.65 (0.47)	28	5.02 (0.52)	28	−2.324	52	0.024	0.322
F3	4.93 (0.75)	28	5.43 (0.49)	28	−2.832	52	0.007	0.392
F4	4.84 (0.66)	28	5.30 (0.53)	28	−2.810	52	0.007	0.389
Quality and Safety in Nursing Care								
2021–2022 academic year								
F1	5.07 (0.38)	14	5.19 (0.28)	14	−0.975	26	n.s	
F2	5.02 (0.53)	14	4.73 (0.42)	14	1.590	26	n.s	
F3	4.61 (0.47)	14	5.52 (0.36)	14	−5.753	26	0.001	1.121
F4	4.66 (0.57)	14	5.04 (0.59)	14	−1.722	26	n.s	
Human Histology (Dentistry)								
2021–2022 academic year								
F1	4.44 (0.57)	17	4.75 (0.50)	17	−1.639	32	n.s	
F2	4.72 (0.54)	17	4.74 (0.57)	17	−0.048	32	n.s	
F3	5.13 (0.75)	17	5.38 (0.41)	17	−1.137	32	n.s	
F4	4.27 (0.56)	17	4.64 (0.48)	17	−2.017	32	0.05	0.356

Range [1 to 6 points]. F1: Competences for treatment in accordance with patient’s age. F2: Competences for treatment in accordance with patient’s sex. F3: Competences for treatment in accordance with patient’s emotional situation. F4: Competences for treatment in accordance with patient’s ethnic-cultural origin.

**Table 5 ijerph-19-14573-t005:** CCI-U questionnaire. Analysis of the level of individual improvement among psychology, nursing and dentistry students in the control and intervention groups, classified by academic year.

	F1		F2		F3		F4	
Subject, Year and Group	Z	*p*	Z	*p*	Z	*p*	Z	*p*
Psychology								
2019–2020 academic year								
Group 1								
Control (*n* = 43)	−1.925	n.s	−0.851	n.s	−0.958	n.s	−0.870	n.s
Intervention (*n* = 44)	−3.175	0.001	−1.520	n.s	−2.067	0.039	−2.366	0.018
Group 2								
Control (*n* = 23)	−1.845	n.s	−0.604	n.s	−0.658	n.s	−0.667	n.s
Intervention (*n* = 22)	−3.242	0.001	−2.936	0.003	−2.264	0.024	−3.032	0.002
2020–2021 academic year								
Group 1								
Control (*n* = 33)	−0.891	n.s	−0.085	n.s	−1.796	n.s	−0.109	n.s
Intervention (*n* = 33)	−2.918	0.004	−2.192	0.028	−1.428	n.s	−3.270	0.001
Group 2								
Control (*n* = 33)	−1.793	n.s	−1.260	n.s	−0.146	n.s	−0.134	n.s
Intervention (*n* = 33)	−1.979	0.048	−1.600	n.s	−3.998	0.001	−3.595	0.001
2021–2022 academic year								
Group 1								
Control (*n* = 22)	−0.072	n.s	−0.095	n.s	−0.642	n.s	−0.243	n.s
Intervention (*n* = 22)	−0.044	n.s	−2.212	0.027	−1.910	0.050	−2.709	0.007
Group 2								
Control (*n* = 14)	−0.411	n.s	−1.118	n.s	−1.118	n.s	−0.760	n.s
Intervention (*n* = 14)	−0.268	n.s	−2.440	0.032	−1.170	n.s	−2.168	0.030
Group 3								
Control (*n* = 17)	−0.161	n.s	−0.457	n.s	−0.599	n.s	−0.158	n.s
Intervention (*n* = 17)	−2.454	0.014	−1.957	0.050	−0.763	n.s	−3.134	0.002
Basic Nursing Methodologies								
2019–2020 academic year								
Control (*n* = 21)	−1.759	n.s	−0.143	n.s	−0.024	n.s	−0.933	n.s
Intervention (*n* = 20)	−0.241	n.s	−1.470	n.s	−0.372	n.s	−2.206	0.027
2020–2021 academic year								
Control (*n* = 22)	−1.273	n.s	−1.400	n.s	−0.110	n.s	−0.286	n.s
Intervention (*n* = 23)	−0.812	n.s	−1.079	n.s	−1.254	n.s	−0.087	n.s
2021–2022 academic year								
Control (*n* = 28)	−1.050	n.s	−1.551	n.s	−1.662	n.s	−0.314	n.s
Intervention (*n* = 28)	−2.454	0.014	−1.957	0.050	−0.763	n.s	−3.134	0.002
Quality and Safety in Nursing Care								
2021–2022 academic year								
Control (*n* = 14)	−1.655	n.s	−0.410	n.s	−1.472	n.s	−0.985	n.s
Intervention (*n* = 14)	−1.460	n.s	−0.631	n.s	−1.927	0.050	−1.981	0.043
Human Histology (Dentistry)								
2021–2022 academic year								
Control (*n* = 17)	−0.726	n.s	−1.025	n.s	−0.040	n.s	−1.921	n.s
Intervention (*n* = 17)	−2.136	0.033	−0.171	n.s	−2.955	0.011	−3.215	0.001

F1: Competences for treatment in accordance with patient’s age. F2: Competences for treatment in accordance with patient’s sex. F3: Competences for treatment in accordance with patient’s emotional situation. F4: Competences for treatment in accordance with patient’s ethnic-cultural origin.

## Data Availability

The data that support the findings of this study are available on request from the corresponding author, M.S.D.M.

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
