# Peer review of "Development, Application and Evaluation of an Active Learning Methodology for Health Science Students, Oriented towards Equity and Cultural Diversity in the Treatment and Care of Geriatric Patients"

_ijerph, 2022, doi:10.3390/ijerph192114573_

Round 1
Reviewer 1 Report
The topic of the article concerning the increasing number of elderly citizens with different needs and requirements is highly topical and important. The article evaluates a new curriculum offered to three different training programs over several years. On a number of parameters it shows by means of a questionnaire that the students have gained more knowledge about how to meet this diverse patient group. However, in the authors' paper there are some ambiguities that require more explanation and thoroughness for the article to be published.
I think as the authors do that it is important to have much more focus on what I would call cultural competencies in health education. However the authors talk about "invisible care" which they define on p. 2 but I don't know if that is a local term. I know the term and recognize the area of care it describes, but why set "invisible care" as a goal for competency development? I would suggest instead that the authors write what knowledge and competencies the students will gain through the new curriculum instead of writing about invisible care, or use the concept of cultural competences ( see e.g. works by Irena Papadopulos) as the collective term for the competences that make invisible care visible.
We are told that the new curriculum is different from the traditional one but we are not given much information about what the traditional curriculum contain other than that it is "technical" which creates an unnecessary polarisation between what students learn now versus the new curriculum. I just don't know what is taught in the Spanish courses and this makes it difficult to relate to the new curriculum, which we also don't get much knowledge about in terms of content. If possible, the authors could provide a link to where the curriculum for traditional education can be found and also provide the new curriculum as supplementary material to the article.
Another element that need to be more clear before the article can be published is the choice of what the authors call "active methodology" - it is not clear what it is and what didactical approaches are drawn upon. The description adds that it includes working with a patient case, which is frequently used in health education programmes I know of. How is it different here? Is it the reflection on the case and what is it driven by ?
Reviewer 2 Report
2.1. Participants -The thesis does not explain why dental students were chosen.The dental students were only in their third year of study, which causes an inhomogeneous sample -in my opinion, the sample of respondents was poorly chosen
Reviewer 3 Report
Section 1.1:
Page 1 – Should be “reside in Spain.”
Throughout the paper, be consistent with the use of “over the age of 65” and “65 and older”, as the former is not inclusive of the age of 65, while the latter is.
Page 2 – Is 2010 in this sentence supposed to be 2020: “the number of people with dementia in Spain has risen, from 113,874 in 2011 to 443,976 in 2010.”
Page 1-2: In the following sentence, it is odd to say that people between the ages of 45 and 64 will soon be joining the 65 and older demographic, as those who are on the younger side of this age grouping will not be entering this demographic for 10-20 years: “These figures warn of a gradual aging of the Spanish population, and it should also be borne in mind that the almost 1.3 million immigrants on the Spanish census who are currently aged between 45 and 64 years will soon be joining this older demographic segment.”
Authors did not make an adequate case that increases in aging immigrants in Spain is linked to increases in the overall proportion of older adults and cases of dementia in the country.
Authors should provide some background on current training among Health Science students, and why new, innovative teaching methods for the provision of theoretical and practical knowledge.
Section 1.2:
Authors should provide some background on the 2030 Agenda for Sustainable Development. Why is this important in the context of this intervention?
What is the source of this fact: “the care of older adults in the health system has mainly been the duty of nursing staff”?
Section 2.1:
What was the racial/ethnic make-up of the intervention and control groups? How might this have impacted the overall results of the study?
Section 2.3:
Provide additional information about the experts who designed the CCI-U questionnaire. What made them experts? In what area(s) are they considered to be experts?
How was the initial list of 100 items created? Were they pulled from other validated tools?
Section 2.4:
Authors should provide additional description of the “non-active traditional methodology” experienced by the control group. What did this entail? Were they exposed to the case study of the geriatric patient Shelima, from the Maghreb, just in a “non-active” way?
Methods, Results, and Discussion Sections:
To what extent did COVID impact the student experiences of the first 2 study years – 2019-2020 and 2020-2021? This is described a little, but additional detail would be beneficial.
Discussion:
To what extent might the racial/ethnic make-up of the groups have impacted the results?
What is the Care-Q Caring Assessment? Does it also measure invisible competencies?
Unless the control group received similar content, just in a “non-active” manner, I’m not sure the authors can state that the active methodology promotes greater awareness in regards to invisible care of the geriatric patient. Authors need to clarify the type of content each group received was similar, it was only the teaching methods that differed.
Round 2
Reviewer 3 Report
The authors adequately addressed my comments/concerns.